# Comparison of the data mining and machine learning algorithms for predicting the final body weight for Romane sheep breed

Cem Tırınk[1]*, Hasan Önder[2], Dominique Francois[3], Didier Marcon[4], Uğur Şen[5], Kymbat Shaikenova[6], Karlygash Omarova[6], Thobela Louis Tyasi[7]

1 Department of Animal Science, Faculty of Agriculture, Igdir University, Iğdır, Türkiye, 2 Department of Animal Science, Faculty of Agriculture, Ondokuz Mayis University, Samsun, Türkiye, 3 GenPhySE, INRA, INPT, INP-ENVT, Université de Toulouse, Castanet Tolosan, France, 4 INRAE, UE P3R, Osmoy, France, 5 Department of Agricultural Biotechnology, Faculty of Agriculture, Ondokuz Mayis University, Samsun, Türkiye, 6 Department of Technology and Processing of Livestock Production, Faculty of Veterinary and Animal Husbandry Technology, Saken Seifullin Kazakh Agrotechnical University, Astana, Kazakhstan, 7 Department of Agricultural Economics and Animal Production, School of Agricultural and Environmental Sciences, University of Limpopo, Sovenga, Limpopo, South Africa

* cem.tirink@gmail.com

**Data Availability Statement:** The data and figures accessible on https://figshare.com/s/3e7db9a7d47071114de6.

## Abstract

The current study aimed to predict final body weight (weight of fourth months of age to select the future reproducers) by using birth weight, birth type, sex, suckling weight, age at suckling weight, weaning weight, age at weaning weight, and age of final body weight for the Romane sheep breed. For this purpose, classification and regression tree (CART), multivariate adaptive regression splines (MARS), and support vector machine regression (SVR) algorithms were used for training (80%) and testing (20%) sets. Different data mining and machine learning algorithms were used to predict final body weight of 393 Romane sheep (238 female and 155 male animals) were used with different artificial intelligence algorithms. The best prediction model was obtained by CART model, both training and testing set. Constructed CART models indicated that sex, suckling weight, weaning weight, age of weaning weight, and age of final weight could be used as an indirect selection measure to get a superior sheep flock on the final body weight of Romane sheep. If genetically established, the Romane sheep whose sex is female, age of final weight is over 142 days, and weaning weight is over 28 kg could be chosen for affording genetic improvement in final body weight. In conclusion, the usage of CART procedure may be worthy of reflection for identifying breed standards and choosing superior sheep for meat yield in France.

## Introduction

Sheep is one of the first domesticated animals within the scope of archaeological and genetic studies [1, 2]. As a result of this domestication process, it has led to an improvement in sheep breeding over time. Multi-purpose sheep breeding is very noteworthy not only for the development of healthy civilizations but also for obtaining animal products such as milk, meat, and

**Funding:** The authors received no specific funding for this work.

**Competing interests:** The authors have declared that no competing interests exist.

fleece, especially in the development processes of rural economies [3–6]. According to this importance for human being, researchers aimed sheep farms to reach a more profitable, innovative, and sustainable [7]. According to the FAO, 5.5 million heads of sheep are adopted to France's several regions [8]. Principal sheep breeds include Charollais, Ile de France, Berrichon du Cher, Blanc du Massif Central, Roussin de La Hague, Préalpes du Sud, Lacaune and Romane in France [9–11]. Sheep breeds suitable for the regions have been adapted for sustainable sheep breeding, along with the optimization of care, food, and other environmental requirements [12].

Knowing the live weights of the animals in the herd in sheep breeding is essential in determining the breeding strategy to be applied in the herd and herd management. In this context, it will provide convenience in calculating the optimum amount of feed per sheep, determining the drug doses, marketing price more reliably, and optimum slaughter time of the animals [13–15].

In literature, there are many studies in which multivariate statistical methods are used together with biometric measurements to create a characterization for breeds [16–19]. Within the framework of the regression analysis assumptions, requirements such as linearity, normal distribution, constant variance, and explanatory variable independence should be sought in multivariate statistical approaches [20–23]. However, various data mining and machine learning algorithms do not need these assumptions [24, 25]. In this context, many studies conducted within the scope of multivariate statistical methods have been carried out on different breeds and species, in which characteristics such as birth weight, live weight, weaning weight, and final weight are tried to be estimated [26–28]. The main purpose of trying to predict such different traits is that these traits vary among animal materials. Expressing these differences will provide convenience in herd management issues such as selection practices and breed characterization.

Using information on sex, birth weight, suckling weight, age of suckling weight, weaning weight, age at weaning weight, and age at final weight, this study attempted to estimate the ideal final weight (weight at the fourth month of age) to choose the future reproducers (rams and ewes)). Three algorithms, including the CART, MARS, and SVR algorithms, were utilized and compared to achieve this goal.

## Materials and methods

Data on the Romane sheep breed was provided by INRA in order to compare the algorithms. Romane sheep (Berrichon du Cher by Romanov crossbred) was developed to increase the productivity of the French sheep herd. INRA created the synthetic strain INRA 401 by crossing the Berrichon du Cher breed (good meat yield and quality but not very prolific, not very maternal, white fleece) with the Romanov breed (very prolific, maternal, but low butchering skills and coloured fleece). The line obtained was named Romane breed in 2006. It has an average litter size of 2 lambs per birth (for 3-year-old ewes), excellent fertility in the off-season, and good viability of the lambs at birth [29]. Housing conditions during the suckling period: ewes and lambs in the barn, ewes are fed with a total mix ration composed of wrapped forage bales and concentrates, and lambs are fed ad-libitum with concentrates. This study used data from 393 animals (155 male and 238 female) born in August, September and October 2021. Sex, birth weight, suckling weight, age at suckling weight (about 30 days), weaning weight, age at weaning weight, and age at final body weight were used to estimate the final weight that weight of fourth months of age to select the future reproducers (rams and ewes), still young and not mature adults.

Descriptive statistics were showed to response and explanatory variables according to each sex. The comparison of each variable was compared by using the independent samples t-test ($p < 0.05$).

The Classification and Regression Tree (CART) algorithm was proposed by Breiman et al. [30]. With the CART algorithm, a binary split tree structure created by splitting a variable homogeneously includes the two sub-nodes. In the CART algorithm, the process begins from the root node, including the initial data set, and continues until many homogeneous sub-nodes are gotten, which will supply the minimum error variance.

Multivariate Adaptive Regression Splines (MARS) algorithm suggested to overcome the classification and regression type problems by Friedman [31]. The MARS algorithm is one of regression procedure that facilitates a more influential description of interaction, linear and nonlinear effects between explanatory and response variables. There is no need for the MARS algorithm assumptions as in linear regression [27, 32–34].

The MARS algorithm generates base functions according to a step-by-step procedure, taking into account all possible interaction effects between candidate nodes and explanatory variables. The algorithm includes two different steps such as forward and backward pass steps, respectively [35]. The initial steps is the forward pass process which begins to determine the term of intercept in the initial pattern, and to improve the model, iteratively contains the initial patterns coupled with the least training error. The forward pass steps typically products an overfitted configuration that achieves extreme complexity [27, 31]. The model built from the forward pass process fits predominantly worthy. However, it can be difficult to overfitting in terms of generalization ability. The initial patterns that specify the smallest amount of the prediction model are abolished in the backward pass process, and this process is hand-me-down in the resolution to this problem [18, 35, 36]. The MARS algorithm is a significant instrument that can take linear and nonlinear relationships between dependent and independent variables [37, 38]. The equation for MARS procedure utilized to predict body weight from explanatory variables is below.

$$\hat{y} = \beta_0 + \sum_{m=1}^{M} \beta_m \prod_{k=1}^{K_m} h_{km}(X_{v(k,m)}) \tag{1}$$

here, $\hat{y}$: the predicted value of BW, $\beta_0$: the intercept of the model, $\beta_m$: the basis functions' coefficient, $K_m$: the parameter that determines the degree of interaction, $h_{km}(X_{v(k,m)})$: the determined basis functions of the model.

Generalized cross-validation error (GCV) is eliminated by using variable selection (forward and backward) and thus the performance of the model is increased. The calculation of GCV is [27, 36, 39]:

$$GCV(\lambda) = \frac{\sum_{i=1}^{n} (y_i - \hat{y}_i)^2}{\left[1 - \frac{M(\lambda)}{n}\right]^2} \tag{2}$$

here; $n$: the size of training set, $y_i$: response variable' observed value (BW), $(\hat{y}_i)$: the estimation of the response variable (BW), $M(\lambda)$: is called the penalty term for the complexity for the model that includes the $\lambda$ terms.

At the initiation of the MARS procedure, the relationship between the explanatory variables, called multicollinearity, was tested and it was determined that there was no multicollinearity problem between the explanatory variables. To predict BW utilization in the training set. The cross-validation procedure helped to select the greatest MARS model among 72 MARS models (degree = 1:2 and nprune = 2:38). In addition, for training data set, ten-fold cross-validation technique was utilized for optimal MARS model.

A significant twig of the support vector machine (SVM), which is a machine learning algorithm, is the support vector regression (SVR) algorithm [40]. Here, struggling with classification is named support vector classification (SVC), while struggling with modeling and prediction is named SVR [33, 41–43]. Although SVR is also a supervised learning method, the prediction performance obtained from SVR varies depending on the training and test set [33, 44].

The main goal in the linear SVR model is to define a function of f(x) which can have the maximum deviation ($\varepsilon$) from the training set. Training set points are built into the boundary between $-\varepsilon$ to $+\varepsilon$ [44]. However, in most studies, it cannot be shown within the scope of linear properties. Therefore, when using nonlinear SVR, the input data is matched to a higher dimensional Hilbert space (H) so that the regression line can be linear [40].

The hyperplane of the nonlinear SVR to be obtained is as follows;

$$\hat{y} = \langle w, \phi(x) \rangle + b \tag{3}$$

In this equation, weight vector is defined by w, non-linear kernel functions are defined by $\phi$(x), indicates vector inner product is defined by $\langle .,. \rangle$ and the term of b is a bias term. There are many nonlinear kernel functions. In the current study, gaussian radial basis kernel function were used.

In comparison of the model performances, the following goodness of fit criteria were used [27, 28, 33, 36, 45]:

1. Root-mean-square error (RMSE):

$$\text{RMSE} = \sqrt{\frac{1}{n} \sum_{i=1}^{n} \left( y_i - y_{ip} \right)^2} \tag{4}$$

2. Akaike information criterion (AIC):

$$\begin{cases} AIC = n.ln\left[ \dfrac{1}{n} \sum_{i=1}^{n} \left( y_i - y_{ip} \right)^2 \right] + 2k, & \text{if } n/k > 40 \\[4mm] AIC_c = AIC + \dfrac{2k(k+1)}{n-k-1} & otherwise \end{cases} \tag{5}$$

3. Standard deviation ratio (SD$_{ratio}$):

$$\text{SD}_{ratio} = \frac{S_m}{S_d} \tag{6}$$

4. Global relative approximation error (RAE):

$$\text{RAE} = \sqrt{\frac{\sum_{i=1}^{n} \left( y_i - y_{ip} \right)^2}{\sum_{i=1}^{n} y_i^{\,2}}} \tag{7}$$

6. Mean absolute percentage error (MAPE):

$$\text{MAPE} = \frac{1}{n} \sum_{i=1}^{n} | \frac{y_i - y_{ip}}{y_i} | *100 \tag{8}$$

7. Pearson correlation coefficient (r):

$$r = \frac{\sum (x_i - \bar{x})(y_i - \bar{y})}{\sqrt{\sum (x_i - \bar{x})^2 \sum (y_i - \bar{y})^2}} \tag{9}$$

8. Performance Index (PI):

$$\text{PI} = \frac{\text{rRMSE}}{1 + r} \tag{10}$$

where, n is the size of training data set, k is called the number of parameters for the model, $y_i$ is the real value of the response variable (BW), $\hat{y}_i$ is the predicted value for response variable (BW), $s_d$ is the standard deviation for the response variable (BW), $s_m$ is the standard deviation for optimum model's errors [33].

RMSE, $SD_{ratio}$, CV, PI, RAE, MAPE and AIC, r and $R^2$ criteria were used to compare the performance of the model. For this aim, it needs to determine best model for smallest RMSE, $SD_{ratio}$, CV, PI, RAE, MAPE and AIC values for train and test set and also, the highest r, $R^2$ value for all algorithms [46].

All statistical evaluation was performed using R software [47]. To required knowledge of the data, descriptive statistics were utilized. The descriptive statistics for explanatory and response variables were predicted by using "psych" package in R environment [48]. The "caret" packages in the R software were used to perform analyze of the CART and MARS algorithms [49]. Also, support vector regression algorithm was performed by using "e1071" package in R software [50]. To display the performances of the constructed all models, the "ehaGoF" package was utilized [51].

## Results

Descriptive statistics for all variables by sex (male and female) factor were expressed as mean ± standard error and are given in Table 1.

In Table 2, Pearson's correlation coefficients for defining the association with response and explanatory variables except for age of suckling weight, age of weaning weight, and age of final weight. Final body weight had a small correlation coefficient with the birth weight, suckling weight, weaning weight with the coefficients of 0.29, 0.42, 0.52, respectively. Among the explanatory variables, relatively high correlation was determined only between weaning weight and suckling weight. The other correlation coefficients were low or moderate and can be interpreted as having no relationship.

Fig 1 points out the constructed CART regression tree diagram in estimation of final weight from explanatory variables such as birth weight (bw), suckling weight (sw), age at suckling weight (asw), weaning weight (ww), age at weaning weight (aww), and age at final weight (afw) and sex (1 = male, 2 = female) factor.

**Table 1. Descriptive statistics for the whole characteristics.**

| Traits | Male | Female |
|---|---|---|
| Birth weight [kg] | 4.39 ± 0.06 | 4.39 ± 0.05 |
| Suckling weight [kg] | 12.66 ± 0.19 | 12.59 ± 0.14 |
| Age of suckling weight [day] | 30.74 ± 0.19 | 31.17 ± 0.13 |
| Weaning weight [kg] | 26.29 ± 0.38 | 24.78 ± 0.27 |
| Age of weaning weight [day] | 64.30 ± 0.24 | 64.66 ± 0.21 |
| Final weight [kg] | 42.16 ± 0.37[a] | 35.75 ± 0.34[b] |
| Age of final weight [kg] | 109.61 ± 0.35 | 105.95 ± 0.83 |

a,b: different letters in same row shows the statistical difference (p<0,05).

The final weight of the Romane sheep breed was divided into two groups based on sex. On the left side of the diagram, if the sex was female, the mean of the final body weight was determined as approximately 36 kg. The female Romane sheep were divided into two groups based on weaning weight, that is, weaning weight <23 kg and weaning weight ≥23 kg. If the weaning weight was ≥23 kg, the tree was divided into the age of final weight<106 days with the mean of 37 kg. In addition, if the age of the final weight was above 106 days, the final weight was divided into two groups as for weaning weight was over and under 27 kg. According to Fig 1, if the sex was female, weaning weight ≥23 kg, age at final weight≥106 days, and weaning weight≥27 kg, the predicted final weight was determined as 45 kg. On the right side of the diagram, if the sex was male, the mean of the final body weight was determined as approximately 42 kg. The male Romane sheep were divided into two groups for the age at final weight at 142 days. If the age of final weight is under 142 days, the final body weight was divided into for suckling weight with the coefficient of 18 kg. In addition, if the sex was male, the age of final weight was over 142 days, weaning weight over 28 kg, the final body weight was determined as 64 kg.

Table 3 gives the results of the CART algorithm depending on the cross-validation procedure. The constructed CART regression tree algorithm made 12 terminal knots (size of regression tree) with relative error (0.148) and the cross-validation error (0.312) which means that coefficient of determination ($R^2$) and cross-validation $R^2$ were close to each other with 0.852 (1–0.148) and 0.688 (1–0.312), respectively (Table 3). The relative error can estimate the $R^2$, and cross-validation error can be defined as the mean of the cross-validated prediction errors and cross-validation $R^2$ is calculated by 1–relative error. Cross-validation standard deviation can be defined as the standard deviation of the cross-validated prediction errors (Table 3).

Table 4 shows that the final body weight can be explained with the eight basic functions in the MARS prediction model. The first term of the MARS prediction model is the intercept with value of 36.770. In the second term, the case where the sex is female for a negative coefficient of -4.369. The third term (Suckling weight-12.8) had a cutpoint of 12.8 kg with a coefficient of 0.814. The fourth term was the suckling weight age, with a cutpoint of 35 with a

**Table 2. Correlation coefficients of the explanatory variables.**

| | Birth weight | Suckling weight | Weaning weight | Final body weight |
|---|---|---|---|---|
| Birth weight | 1 | | | |
| Suckling weight | 0.68 | 1 | | |
| Weaning weight | 0.53 | 0.79 | 1 | |
| Final body weight | 0.29 | 0.42 | 0.52 | 1 |

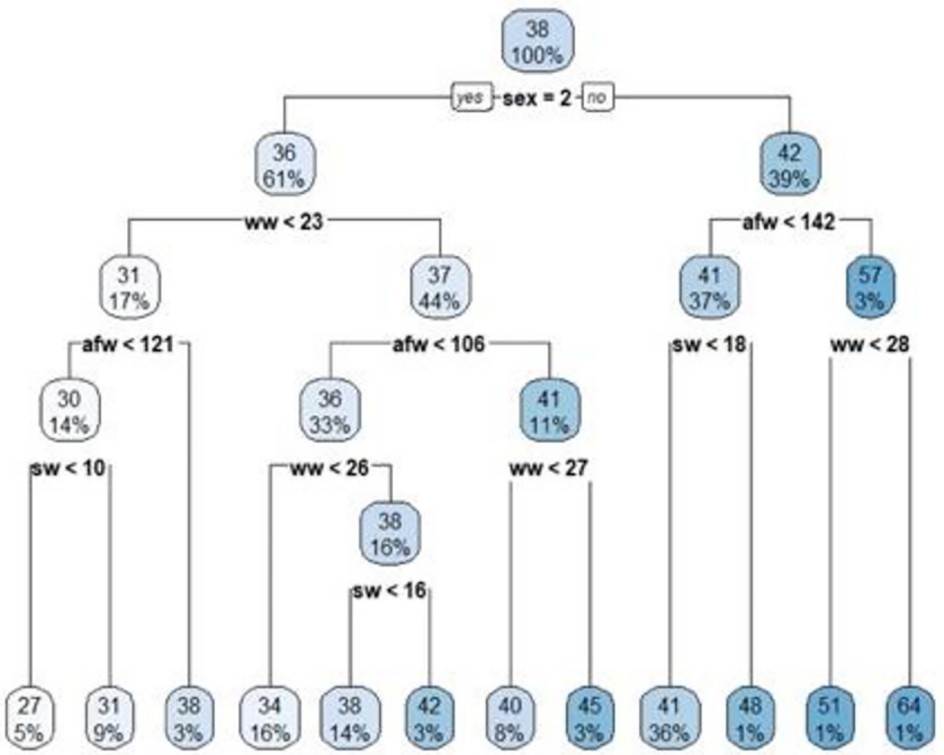

**Fig 1. Constructed CART regression tree diagram.**

coefficient of 0.415. For the age of suckling weight is 35 days, the effect of term four on final body weight was unavailable. The fifth and sixth terms were for weaning weight, with cutpoint 17.4 and 24.4 with a coefficient of 1.0817 and -0.616, respectively (Table 4). When weaning weight was 17.4 and 24.4 kg or lighter, no contribution of the weaning weight to the final body weight was found. For the weaning weight lighter than 17.4 kg, term 5 was useless, and then only the term 6 (Weaning weight-24.4) contributed to the final body weight. The same procedure was for the term of 6. If the weaning weight was lighter than 24.4 kg, the term of 6 was

**Table 3. Results of CART algorithm based on the cross-validation technique.**

|  | Complexity parameter | Number of splits | Relative error | Cross-validation error | Cross-validation standard deviation |
|---|---|---|---|---|---|
| 1 | 0.294 | 0.000 | 1.000 | 1.014 | 0.138 |
| 2 | 0.177 | 1.000 | 0.706 | 0.759 | 0.103 |
| 3 | 0.138 | 2.000 | 0.530 | 0.675 | 0.096 |
| 4 | 0.067 | 3.000 | 0.391 | 0.451 | 0.050 |
| 5 | 0.046 | 4.000 | 0.325 | 0.399 | 0.048 |
| 6 | 0.042 | 5.000 | 0.279 | 0.384 | 0.046 |
| 7 | 0.028 | 6.000 | 0.237 | 0.318 | 0.042 |
| 8 | 0.017 | 7.000 | 0.208 | 0.320 | 0.045 |
| 9 | 0.015 | 8.000 | 0.191 | 0.322 | 0.045 |
| 10 | 0.015 | 9.000 | 0.176 | 0.323 | 0.046 |
| 11 | 0.013 | 10.000 | 0.161 | 0.311 | 0.045 |
| 12 | 0.010 | 11.000 | 0.148 | 0.312 | 0.045 |

**Table 4. The obtained MARS model for Romane sheep.**

| Explanatory variables | Coefficients |
| --- | --- |
| Intercept | 36.770 |
| Sex-Female | -4.369 |
| Suckling weight-12.8 | 0.814 |
| 35-Age of suckling wight | 0.415 |
| Weaning weight-17.4 | 1.0817 |
| Weaning weight-24.4 | -0.616 |
| 133-Age of final weight | -0.222 |
| Age of final weight-133 | 0.324 |

useless. The seventh and the last term of the MARS prediction model were for the age of final weight with a cutpoint of 133 days.

To estimate the final body weight, the obtained best prediction MARS model allows breeders to make more precise decisions considering herd management, such as the necessary feed amount, medicinal drug doses for Romane sheep breed, and establishing the selling value of each sheep.

At the start of the SVR algorithm was instructed to the training set. Later performing the training processes, the SVR was assessed to predict the final body weight for Romane sheep breed. In the current study, the gaussian radial basis kernel function was used for estimating the final body weight of the Romane sheep breed. The consistency of the model is based on the selection parameters such as cost (C), gamma and epsilon. The current study determined cost, gamma, and epsilon values as 1, 0.083, and 0.1, respectively. These parameters were verified for several values, and analysis was affected for C and epsilon values. So, it would provide the greatest trustworthy model. To determine which variable is more effective, sensitivity analysis was carried out. The sensitivity analysis was calculated to estimate the relative importance values of the affected explanatory variables on final body weight. In Fig 2, the sensitivity analysis results were given.

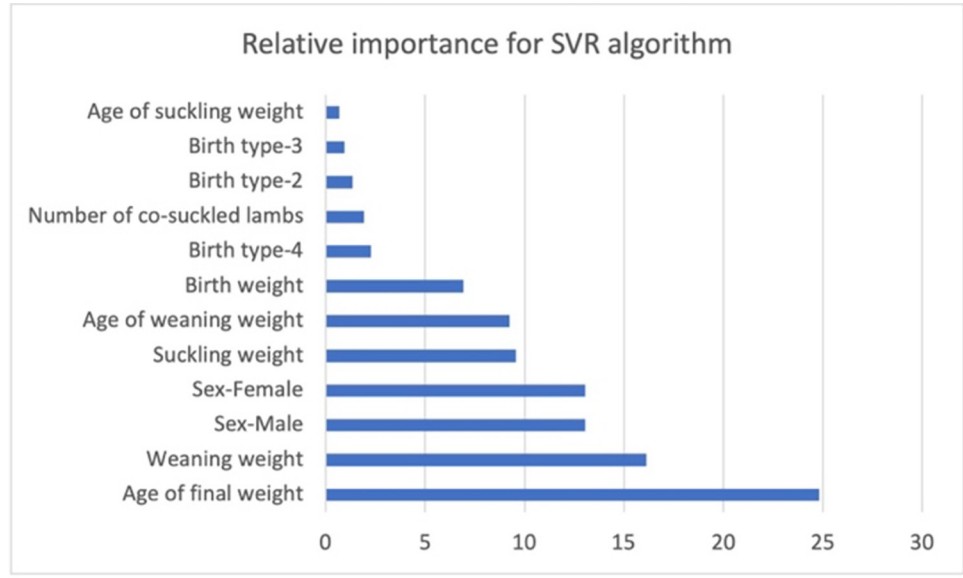

**Fig 2. Sensitivity analysis for support vector regression algorithm.**

**Table 5. Goodness-of-fit criteria for CART, MARS and SVR algorithms.**

| Goodness-of-fit criteria | CART | | MARS | | SVR | |
|---|---|---|---|---|---|---|
| | **Train** | **Test** | **Train** | **Test** | **Train** | **Test** |
| RMSE | 2.264 | 2.586 | 2.466 | 2.867 | 2.404 | 2.898 |
| $SD_{ratio}$ | 0.385 | 0.434 | 0.419 | 0.483 | 0.409 | 0.489 |
| CV | 5.920 | 6.790 | 6.450 | 7.550 | 6.280 | 7.640 |
| r | 0.923 | 0.902 | 0.908 | 0.876 | 0.924 | 0.882 |
| PI | 3.075 | 3.563 | 3.375 | 4.004 | 3.263 | 4.034 |
| RAE | 0.003 | 0.004 | 0.004 | 0.006 | 0.004 | 0.006 |
| MAPE | 4.856 | 5.596 | 5.022 | 5.484 | 4.162 | 5.773 |
| $R^2$ | 0.852 | 0.810 | 0.824 | 0.766 | 0.833 | 0.761 |
| AIC | 516.504 | 146.321 | 586.353 | 178.198 | 554.251 | 163.863 |

According to Fig 2, the age of final weight was a more effective variable on final body weight for support vector regression algorithms. The second effective variable was weaning weight. Apart from these variables, sex variables (male and female), suckling weight, age of weaning weight, and birth weight were also important in determining the final body weight. The least effective variables in determining the final body weight were the age of suckling weight, birth type (2, 3, and 4), and number of co-suckled lambs.

For comparing all models, the goodness of fit criteria was used. The model performance results for CART, MARS, and SVR algorithms, based on the goodness-of-fit criteria, were provided in Table 5. Table 5 shows the finest prognostic model was realized for CART algorithm. The CART algorithm had the smallest RMSE, $SD_{ratio}$, CV, PI, RAE, MAPE, and AIC values for the train and testing set. Also, the highest $R^2$ value was determined for the CART algorithm.

In addition, The Pearson correlation coefficient was determined as 0.923 for the original data and 0.902 for the training and test sets for the predicted data, respectively. In addition, since the coefficient of variation (CV) values for both the train and the testing set in each model are below 30%, it is concluded that the results obtained from the applied models are reliable.

## Discussion

Various statistical methods can be used to explain the relationship between body weight and various characteristics. The important point here is to use the correct statistical method. There are even some studies to estimate live weight in different breeds; no study has been found for the Romane sheep breed. In the present study, the final weight of male Romane sheep breed had a greater mean with comparison female Romane sheep breed (p<0.05).

Alonso et al. [43] utilized the SVR algorithm to predict the carcass weight in Asturiana de los Valles beef. To predict the carcass weight, the study was made by using 390 measurements from 144 animals. According to the study's results, 150 days before slaughter time the ideal carcass weight prediction was obtained. They found MAPE values as 4.12 and 4.91 for train and test, respectively. The results of the MAPE values were smaller than our results. The achievement of their study may be caused by a sample size greater than ours.

Celik et al. [3] aimed to compare CART, CHAID, Exhaustive CHAID, MARS, MLP, and RBF for Mengali rams. The greatest prediction model was described as the CART algorithm in the scope of the model comparison criteria such as $R^2$, RMSE and $SD_{ratio}$. The CART and MARS algorithms we used in our study were also used in this study. Celik et al. found that the CART algorithm was more reliable than the MARS algorithm [3]. This result supports our current study results.

Iqbal et al. [52] was aimed to examine performances of the random forests, regression tree, SVR algorithm, and gradient boosting machine algorithms. For this aim, Iqbal et al. [52] was used the data obtained from Beetal goats. To predict the body weight, they some biometric measures such as sex, and body length. As model comparison criteria such as Pearson's correlation, $R^2$, RMSE, MAPE, and MAE were used. The results of Iqbal et al. [52] showed that the gradient boosting machine was stated to be the greatest model for estimating the body weight of Beetal goats. However, the random forest regression algorithm was determined to be the second-best algorithm. Even if the SVR algorithm does not give the best results, even the data structure impacts on the prediction performances.

Marco et al. [53] tried to use AdaBoost ensemble learning method and RFR for different data sets. Several machine learning algorithms, for different data sets MLP, SVR, CART, kNN, and RFR were applied. According to the results obtained from most of the datasets applied in the study, they stated that it is a reliable and successful algorithm for RFR. However they indicated that the CART algorithm was also a useable algorithm, even if the CART algorithm wasn't very sensitive to parameter tuning when RFR was so sensitive to parameter tuning, which results in stable prediction performance.

Tırınk [54] compared various artificial intelligence methods such as Multivariate Adaptive Regression Splines, Random Forest Regression, Bayesian Regularized Neural Network, and Support Vector Regression algorithms to estimate body weight from biometric measurements for the Thalli sheep breed. For this aim, 270 female Thalli sheep breeds were used. According to the results, the MARS algorithm was the best prediction model for Thalli sheep breeds inside the bayesian regularized neural network, random forest regression and support vector regression. These results support our results because the MARS model showed more consistent results than the SVR algorithm.

Tırınk et al. [33] compared CART, SVR, and RFR artificial intelligence methods using body measurements to estimate body weight at a different share of Polish Merino in the genotype of crossbreds (share of Suffolk and Polish Merino genotypes). To compare the estimation performances of the evaluated algorithms and determine the best model for estimating body weight, various body measurements and sex and birth type characteristics were assessed. According to test sets results, using random forest regression was recommended instead of using CART and SVR algorithms. The performance of the SVR ($R^2$ = 0.714) algorithm was found to be more reliable than CART ($R^2$ = 0.578) algorithm by Tırınk et al. [33]. These findings were not compatible with our results. In our study, the CART ($R^2$ = 0.810) algorithm was better than the SVR ($R^2$ = 0.761) algorithm. But it should be considered that there were such big differences in the point of determination coefficient. They used the data for 344 animals, while our sample size was 393. Although these sample sizes were similar, this cannot be the reason for the differences between the algorithms. The structure of the data may be the reason for these differences.

Kumar and Kumar-Singh [55] aimed to compare MLR, MARS, SVR, and RFR techniques in hydrological time-series modelling. Their results showed that according to the RMSE and $R^2$ values, they reported that the SVR algorithm was superior and stated that it was applicable to predict the weekly pan evaporation values for the Ranichauri region. Their results were comparable only for MARS and SVR algorithms with our results. They suggested the use of SVR, but our results had a different recommendation to use MARS algorithms when compared with the study of Kumar and Kumar-Singh [55]. The coefficient of determination difference was only 0.01, which can be ignored that both MARS and SVR can be interpreted as similar, as in our results.

Komadja et al. [56] aimed a model development to predict peak particle velocity (PPV) in opencast mines using CART, MARS, and SVR algorithms. The models were developed using a

record of 1001 real blast-induced ground vibrations, with ten corresponding blasting parameters from 34 opencast mines/quarries from India and Benin. The results showed that the MARS model outperformed other models in this study with lower error (RMSE = 0.227) and $R^2$ of 0.951, followed by SVR ($R^2$ = 0.87), CART ($R^2$ = 0.74), and empirical predictors. Their results completely contrast to our findings that our study showed the CART algorithm was the best. Also, Komadja et al. [56] gave 50 results of previous studies results in a list. Interpretation of the list showed no consensus on the algorithm selection. The contrast in the results may be caused by data type and sample size.

Compared to the abovementioned research, artificial intelligence was used for many species and breeds. The extensive variation in previous studies was attributed to the physiologic phase of the animals, raising systems differences, and selection of statistical methods to apply. When our study was compared with other results, it was determined when the selected goodness-of-fit criteria are examined, it is understood that the models used in this study give similar results. However, proposing some statistical methods for BW estimation using biometric traits is very important for animal production, characterization and breeding purposes. The results obtained showed that much more work needs to be done on this subject.

## Conclusion

Finally, in order to provide breeders and researchers with access to a superior population of Romane sheep for use in upcoming research, the CART method should be recommended. The findings of the current study, which uses the goodness of fit criteria to choose the best model for both CART and other methods, demonstrated that data mining and machine learning algorithms can be successfully used to estimate body weight based on other explanatory variables obtained. Even if there are some variations brought on by the breed factor, as was mentioned in the discussion section, more accurate models can be created by carrying out comparable research in addition to using alternative algorithms.

## Author Contributions

**Conceptualization:** Cem Tırınk, Hasan Önder.

**Data curation:** Cem Tırınk, Hasan Önder.

**Formal analysis:** Cem Tırınk, Hasan Önder.

**Methodology:** Uğur Şen, Kymbat Shaikenova, Karlygash Omarova, Thobela Louis Tyasi.

**Resources:** Dominique Francois, Didier Marcon.

**Software:** Cem Tırınk, Hasan Önder.

**Visualization:** Cem Tırınk.

**Writing – original draft:** Cem Tırınk, Hasan Önder, Dominique Francois, Uğur Şen, Kymbat Shaikenova, Karlygash Omarova, Thobela Louis Tyasi.

**Writing – review & editing:** Cem Tırınk, Hasan Önder, Dominique Francois, Uğur Şen, Kymbat Shaikenova, Karlygash Omarova.

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
