## [Decision Letter · Decision Letter 0]

22 May 2023

PONE-D-23-13209Comparison of the Data Mining and Machine Learning Algorithms for Predicting the Final Body Weight for Romane Sheep BreedPLOS ONE

Dear Dr. Tırınk,

Thank you for submitting your manuscript to PLOS ONE. After careful consideration, we feel that it has merit but does not fully meet PLOS ONE’s publication criteria as it currently stands. Therefore, we invite you to submit a revised version of the manuscript that addresses the points raised during the review process.

We look forward to receiving your revised manuscript.

Kind regards,

yasin altay, Ph. D.

Academic Editor

PLOS ONE

Journal Requirements:

https://www.mdpi.com/2076-2615/13/5/798

https://www.researchgate.net/publication/248164965_Artificial_neural_network_for_prediction_and_control_of_blasting_vibrations_in_Assiut_Egypt_limestone_quarry

www.largeanimalreview.com

In your revision ensure you cite all your sources (including your own works), and quote or rephrase any duplicated text outside the methods section. Further consideration is dependent on these concerns being addressed.

● A clean copy of the edited manuscript (uploaded as the new *manuscript* file).

4. Please note that PLOS ONE has specific guidelines on code sharing for submissions in which author-generated code underpins the findings in the manuscript. In these cases, all author-generated code must be made available without restrictions upon publication of the work. Please review our guidelines at https://journals.plos.org/plosone/s/materials-and-software-sharing#loc-sharing-code and ensure that your code is shared in a way that follows best practice and facilitates reproducibility and reuse.

"There is no financial disclosure."

6. Thank you for stating the following in your Competing Interests section:  

"There is no competing interest."

7. In your Data Availability statement, you have not specified where the minimal data set underlying the results described in your manuscript can be found. PLOS defines a study's minimal data set as the underlying data used to reach the conclusions drawn in the manuscript and any additional data required to replicate the reported study findings in their entirety. All PLOS journals require that the minimal data set be made fully available. For more information about our data policy, please see http://journals.plos.org/plosone/s/data-availability.

8. Please upload a copy of Supporting Information Figure/Table/etc. S1 and S2 File which you refer to in your text on page 15.

Reviewers' comments:

Reviewer's Responses to Questions

**Comments to the Author**

1. Is the manuscript technically sound, and do the data support the conclusions?

Reviewer #1: Yes

Reviewer #2: Partly

Reviewer #3: Partly

2. Has the statistical analysis been performed appropriately and rigorously? 

Reviewer #1: Yes

Reviewer #2: Yes

Reviewer #3: No

3. Have the authors made all data underlying the findings in their manuscript fully available?

Reviewer #1: No

Reviewer #2: Yes

Reviewer #3: Yes

4. Is the manuscript presented in an intelligible fashion and written in standard English?

Reviewer #1: Yes

Reviewer #2: No

Reviewer #3: No

5. Review Comments to the Author

Reviewer #1: The aim of the manuscript is to estimate the optimal final body weight of Romane lambs from specific parameters (birth weight, suckling weight, age at suckling weight, weaning weight, age at weaning weight and age at final weight) using traditional statistical calculations and different data mining and machine learning algorithms (CART, MARS and SVR). This study will help sheep farmers to choose the right selection criteria and flock management, which is an important economic issue.

The strongest part of the manuscript is the statistical analysis, especially the textual evaluation of the results obtained. However, the introduction is very general. It is recommended that the problem statement be more specific.

Formally, the following numbered lines should be corrected:

46 Im-portance

204 Invalid table title

250 It is recommended to delete the letters 'h' from the table.

As there are so many abbreviations, it is recommended to summarise them in a list. It takes a lot of effort to find the abbreviations when reading the analysis.

It would also be useful to indicate the abbreviations in Figure 1 in the text (ww, sw, fw, etc.).

The use of units of measurement in tables is also recommended.

Overall, the manuscript is suitable for publication with the suggested changes.

Reviewer #2: Interesting article with adequate statistical analysis. Document writing deserves to be improved to a standard english.

Comments:

Lina 51: Update breeds references with more recent articles. Those published are between 2003 and 2006

Line 70: Sex (corret sex)

Line 80: Age of suckling weight: Two concepts, Weight and Age??

Suckling Weight or Weight at 30 days?

Line 85: age of weaning weight or weaning age?

Change on all document

Line 115: Add References 37,38 instead of 37, 38

Line 187: Objectify the term "Caret"

Line 195: Traits Use Units (kg) (days)

Line 278 - Discussion - Improve the english

Line 293, 309, 316, 324, 338, 346: Author aimed to compare?

Line 364 - Conclusion It should be more objective, especially in the opening paragraphs. At this point it discusses methodologies

Reviewer #3: The study should be revised in terms of writing language and especially the conclusion part should be summarized a little more. The discussion part of the article should be written using more relevant studies and unnecessary sources should be removed. There are also some expression and/or statistical errors. These must be reviewed from the very beginning.

In general, the study requires a great deal of correction, especially in terms of language.

6. PLOS authors have the option to publish the peer review history of their article (what does this mean?). If published, this will include your full peer review and any attached files.

Reviewer #1: No

Reviewer #2: No

Reviewer #3: No

<quillbot-extension-portal></quillbot-extension-portal>

---

## [Author Response · Author response to Decision Letter 0]

23 Jun 2023

Dear Editor,

Many thanks for sharing valuable comments of you and reviewers with us on improving the manuscript PONE-D-23-13209 entitled “Comparison of the Data Mining and Machine Learning Algorithms for Predicting the Final Body Weight for Romane Sheep Breed”. I have given answers to all comments of the reviewers evaluating my manuscript using red color fonts in the attachment. Also, red color font on the revised manuscript has been used for indicating all the corrections made by Reviewers.

With Best Regards

Assoc. Prof. Dr. Cem TIRINK

Corresponding author

---

## [Editor Report · Decision Letter 1]

2 Jul 2023

PONE-D-23-13209R1Comparison of the Data Mining and Machine Learning Algorithms for Predicting the Final Body Weight for Romane Sheep BreedPLOS ONE

Dear Dr. Tırınk,

Thank you for submitting your manuscript to PLOS ONE. After careful consideration, we feel that it has merit but does not fully meet PLOS ONE’s publication criteria as it currently stands. Therefore, we invite you to submit a revised version of the manuscript that addresses the points raised during the review process.

We look forward to receiving your revised manuscript.

Kind regards,

yasin altay, Ph. D.

Academic Editor

PLOS ONE

Journal Requirements:

Additional Editor Comments:

It is beneficial to invite the reviewers, again to check the arrangements you have made.

Reviewers' comments:

While revising your submission, please upload your figure files to the Preflight Analysis and Conversion Engine (PACE) digital diagnostic tool, https://pacev2.apexcovantage.com/. PACE helps ensure that figures meet PLOS requirements. To use PACE, you must first register as a user. Registration is free. Then, login and navigate to the UPLOAD tab, where you will find detailed instructions on how to use the tool. If you encounter any issues or have any questions when using PACE, please email PLOS at figures@plos.org. Please note that Supporting Information files do not need this step.<quillbot-extension-portal></quillbot-extension-portal>

---

## [Author Response · Author response to Decision Letter 1]

5 Jul 2023

Dear Reviewers,

The citations in the manuscript were reviewed one by one. In addition, the places and requirements in the text were reviewed and it was determined that the citations in the text and in the references section were consistent.

With all respect.

---

## [Editor Report · Decision Letter 2]

18 Jul 2023

Comparison of the Data Mining and Machine Learning Algorithms for Predicting the Final Body Weight for Romane Sheep Breed

PONE-D-23-13209R2

Dear Dr. Tırınk,

We’re pleased to inform you that your manuscript has been judged scientifically suitable for publication and will be formally accepted for publication once it meets all outstanding technical requirements.

Kind regards,

Yasin ALTAY, Ph. D.

Academic Editor

PLOS ONE

Additional Editor Comments (optional):

Dear Authors;

Thank you for appreciating your valuable study on PLOS ONE. The study can be accepted considering that it will contribute positively to the literature.

Reviewers' comments:

<quillbot-extension-portal></quillbot-extension-portal>

---

## [Editor Report · Acceptance letter]

26 Jul 2023

PONE-D-23-13209R2 

Comparison of the Data Mining and Machine Learning Algorithms for Predicting the Final Body Weight for Romane Sheep Breed 

Dear Dr. Tırınk:

I'm pleased to inform you that your manuscript has been deemed suitable for publication in PLOS ONE. Congratulations! Your manuscript is now with our production department. 

Kind regards, 

on behalf of

Dr Yasin ALTAY 

Academic Editor

PLOS ONE